# Evaporation Dynamics of Sessile and Suspended Almost-Spherical Droplets from a Biphilic Surface

Elena Starinskaya [1,2,*], Nikolay Miskiv [1,2], Vladimir Terekhov [1], Alexey Safonov [1], Yupeng Li [3], Ming-Kai Lei [3] and Sergey Starinskiy [1,2]

1    S.S. Kutateladze Institute of Thermophysics SB RAS, Lavrentyev Ave. 1, 630090 Novosibirsk, Russia
2    Novosibirsk State University, Pirogova Str. 2, 630090 Novosibirsk, Russia
3    School of Materials Science and Engineering, Dalian University of Technology, Dalian 116024, China
*    Correspondence: prefous-lm@yandex.ru

**Abstract:** Research in the field of the evaporation of liquid droplets placed on surfaces with special wetting properties such as biphilic surfaces is of great importance. This paper presents the results of an experimental study of the heat and mass transfer of a water droplet during its evaporation depending on the direction of the gravitational force. A special technique was developed to create unique substrates, which were used to physically simulate the interaction of liquid droplets with the surface at any angle of inclination to the horizontal. It was found that the suspended and sessile droplets exhibited fundamentally different evaporation dynamics. It was shown that the suspended droplets had a higher temperature and, at the same time, evaporated almost 30% faster.

**Keywords:** droplet; biphilic surface; evaporation; sessile; pendant; heat and mass transfer

## 1. Introduction

The evaporation of droplets from solid surfaces is an important fundamental process used in various applications such as inkjet printing [1], the controlled deposition of self-assembled surface coatings [2], DNA extraction [3], disease diagnosis and drug development [4,5], spray painting and surface coating with solid particles [6,7], the creation of self-cleaning and water-repellent surfaces [8], removing condensed droplets from rear-view mirrors or windshields [9], increasing yields through the effective application of foliar fertilizers [10], and surface cooling due to a phase transition [11–21].

Many studies have been devoted to the investigation of droplet evaporation on heated surfaces [22–25]. Tran et al. [26] studied the behavior of droplets upon impact with superheated surfaces, noting that, at surface temperatures above the boiling point of the liquid, droplet evaporation differs greatly from natural evaporation on a substrate under normal atmospheric conditions [26]. Gao et al. [9] experimentally studied the evaporation of sessile droplets of various volumes on heated hydrophilic and hydrophobic surfaces under constant heat fluxes. At low heat fluxes, the droplet size strongly affected the evaporation time and this effect decreased with increasing heat fluxes. Many researchers carried out their experiments under normal atmospheric conditions [8,27–34].

The dynamics of droplet evaporation from a solid surface depend on many factors including wettability, evaporation flux at the interface and triple line, substrate temperature, external fields, and thermocapillarity [35,36]. In terms of heat transfer, this process is a complex interaction among convection in the gas and liquid phases, evaporation at the contact line, vapor diffusion, cooling at the liquid–gas interface, and possible Marangoni effects [27]. The droplet evaporation process is very complex and although significant progress has been made, this process is still not fully understood, in particular, with regard to coupled heat transfer near the triple contact line [37]. Droplet evaporation can occur in a constant contact angle mode, as shown in [38]. In some cases, the evaporation of droplets occurs in a mixed mode with a simultaneous change in the contact angle and radius of

the contact line [39]. Yu et al. [28] and Fang et al. [29] observed that the evaporation of droplets on a hydrophobic surface first occurred in a constant contact radius mode with a change in the contact angle (CCR pinning mode) and the subsequent stage was dominated by a constant contact angle mode with a change in the contact area of the droplet with the surface (CCA depinning mode). Subsequently, Yu et al. [40] studied sessile water droplets evaporating on PDMS and Teflon surfaces and found that in all experiments on hydrophobic surfaces, evaporation started with the CCR mode, switched (after a short time) to the CCA mode, and ended with a mixed mode. Most researchers came to the same conclusion, that is, evaporation on hydrophobic surfaces occurs in two stages: in the initial stage, the pinning mode occurs and then the depinning mode prevails [28,29]. However, different results have also been reported in the literature. It has been shown [40,41] that during evaporation, the depinning mode prevails on hydrophobic surfaces and the pinning mode occurs on hydrophilic surfaces. Birdi and Vu [42] and Uno et al. [30] found that droplet evaporation occurs in the CCR mode on a hydrophilic surface and the CCA mode on a hydrophobic surface. In addition, Shin et al. [8] investigated water evaporation on various wetted surfaces and found that both the contact angle and contact area change at the end of droplet evaporation.

It is of great interest to study the evaporation of liquid droplets placed on surfaces whose structures can lead to significant changes in the droplet evaporation process. Such surfaces include biphilic surfaces, which have regions with different wetting properties [43,44]. Over the past few years, a number of studies have shown that micro- and nanostructured surfaces can increase the overall heat transfer coefficient, e.g., in pool boiling [45], flow boiling [46], and film boiling [47]. At the same time, although the study of the role of structured surfaces may be important, there are still many unresolved issues related to fluid flow and heat transfer, even when pure fluids come into contact with smooth surfaces [11]. However, the investigation of the evaporation process of a suspended droplet on a superhydrophobic surface is experimentally very difficult due to the limitation of droplet spatial stabilization. The fixation of a droplet in a specific place can be performed on a biphilic surface. Creating a superhydrophobic surface with a hydrophilic region that will hold the droplet in place and prevent it from rolling will help to solve the problem of evaporation on surfaces with a high contact angle. We have not found similar studies in the literature, but they could be very useful for controlling deposition processes, understanding droplet evaporation processes on surfaces with high adhesion depending on the gravitational fields, designing devices based on droplet evaporation, etc. The sharper the spatial transition from a hydrophilic to a hydrophobic surface, the more effects can be observed. From this point of view, the most attractive surfaces should have a micron-scale transition area from the superhydrophobic to the superhydrophilic region. These surfaces can be used to create microdevices in optics (e.g., plasmonic reflectors) and chemical sensors and biosensorics (substrates for surface-enhanced Raman spectroscopy, SERS).

The intensification of heat transfer, control of fluid movement, etc., were considered in some papers [48]. The evaporation of droplets has also received attention in the literature but there are still very few publications on this topic, e.g., [43,49]. Therefore, at present, it is not clear how evaporation from a material with high adhesion occurs. This study seeks to expand our understanding of the evaporation of a liquid droplet from a biphilic surface. Along with the case where a sessile droplet is located at the point of contact (point of attachment of the droplet to the surface/seat), we also consider the case where the droplet is inverted (suspended). It has been found that the special structure of the surface allows the evaporation of a suspended droplet simultaneously in the CCR and CCA modes. On this surface, a suspended droplet evaporates 30% faster than on a hydrophobic surface. The reasons for this, which are related to the heat and mass transfer processes, are discussed later in this paper.

## 2. Materials and Methods

### 2.1. Preparation of Materials

The creation of unique biphilic substrates with seat points was based on a combination of laser processing and hot-wire chemical vapor deposition (HW CVD) [50,51]. The superhydrophilicity effect was achieved on the surface of single-crystal silicon using laser irradiation. For this, an $8 \times 18$ mm² silicon wafer was treated with the fundamental harmonic of a Nd:YAG laser with a wavelength of 1064 nm, a pulse duration of 9 ns, and a Gaussian spatial profile. The laser beam scanned the substrate surface at a speed of 2 mm/s and the treated area was $10 \times 10$ mm². To achieve the superhydrophilicity effect, the silicon surface was irradiated with 40,000 laser pulses at a frequency of 5 Hz. Next, a fluoropolymer coating was applied to the obtained superhydrophobic surface by HW CVD. The method involved the activation of the precursor gas flow with a hot (670 °C) Nichrome wire catalyst, followed by the precipitation and polymerization of the free radicals formed on the substrate surface. Hexafluoropropylene oxide was used as the precursor gas. The initial surface contact angle of the fluoropolymer under the selected deposition conditions (i.e., without laser removal) was $155 \pm 3°$. The landing seats for droplet fixing were created in the last stage. For this, the resulting sample with fluoropolymer coating was locally (pointwise) irradiated with laser pulses. Round seats of various sizes were obtained by varying the number of laser pulses, radiation focusing, and beam energy (Figure 1a).

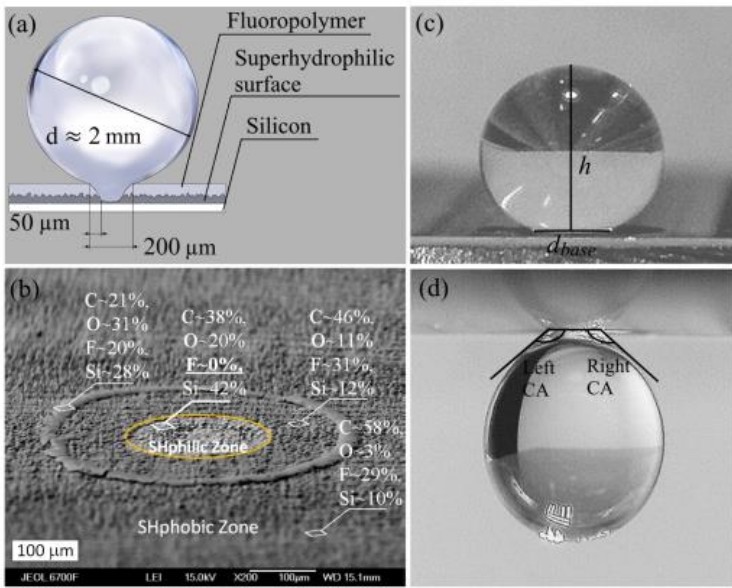

**Figure 1.** (**a**) Schematic of the substrate. (**b**) SEM image of the seat after the evaporation of colloidal solution droplets of SiO₂ particles in distilled water. The indicated compositions in the corresponding square regions were determined by the EDX method. (**c**) Optical image of a sessile droplet in the seat. (**d**) Optical image of a droplet in the seat. $h$, $d_{base}$, Left CA, and Right CA are the droplet height, base diameter, and left and right contact angles, respectively.

The layout of the seat is shown in Figure 1a. The thickness of the wafers was 0.5 mm and the thermal conductivity of the silicon wafer was 149 W/(m·K). The contribution of the Teflon layer can be neglected due to its very thin thickness ($\sim$50 nm). Figure 1b shows an SEM image with the seat and the annular coffee ring deposit obtained by the deposition of solid particles on the contact line during the evaporation of a 0.1 wt % SiO₂ nanofluid sessile droplet. It should be noted that the size of the "coffee ring" corresponds to the measured diameter of the base of the droplet. The obtained biphilic surfaces with seats allowed us to carry out experiments to study the heat and mass transfer processes during droplet evaporation by varying the orientation of the droplet relative to the gravitational forces (Figure 1c,d).

### 2.2. Experimental Setup

The study of the heat and mass transfer processes during the evaporation of the droplets was carried out on the experimental setup shown schematically in Figure 2. The experimental setup consisted of a copper plate to which the substrate was attached and a rotary mechanism. The substrate was attached to the copper surface using thermal paste, then, the required angle was set and the droplet was placed on the seat. The conditions around the droplet were measured with an AZ Instrument model 872 hygrometer to ensure the control of the ambient humidity and temperature with an uncertainty of $\pm 4\%$.

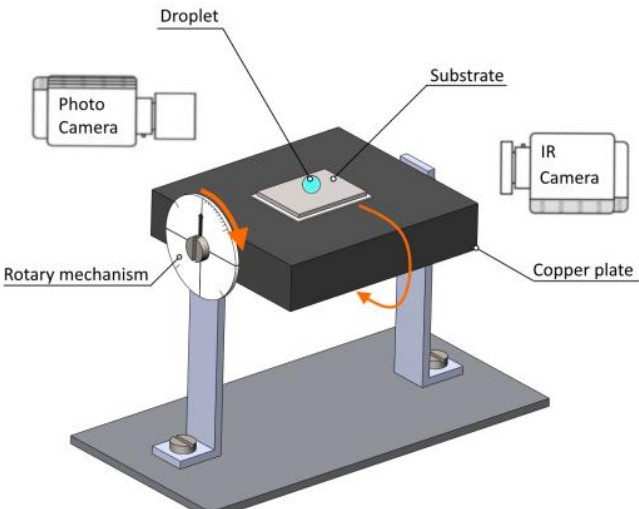

**Figure 2.** Schematic of the experimental setup.

We studied droplets of distilled water with an initial volume of about 6 μL. The ambient temperature and relative humidity corresponded to room-temperature conditions and were 25 °C and $\varphi$ = 21%, respectively. Droplets were formed on the substrate surface using a Thermo Scientific mechanical pipette with an accuracy of ~0.1 μL. The temperature distribution on the droplet surface was determined by infrared thermography (Figure 3a,b). The measurements were carried out with an NEC TH7102IR thermal imaging camera at wavelengths of $\lambda$ = 8–14 μm using a TH 71-377 macro lens. Thermal images were averaged over the area near the central part of the thermal image of the droplet, as shown in Figure 3a,b. The obtained data were processed by the ThermoTracer program. The systematic and random errors of these measurements were estimated at 0.2 °C and 0.45 °C, respectively, resulting in a total error of 0.65 °C. During the experiment, the droplet shape was recorded with a Digi Scope II v3 digital microscope. The contact angle, contact line size, and droplet height were measured from the photographs as shown in Figure 1c,d. The contact angle was measured on the left and right sides (see Figure 1d), and the obtained values were then averaged. Each evaporation experiment was repeated 3 times and good reproducibility of all geometric parameters was observed. The error in the determination of the geometric parameters was the sum of the instrumental error (0.05 mm) and the statistical error (0.02 mm). The change in the shape of the evaporating droplets is shown in the photographs (Figure 3c,d).

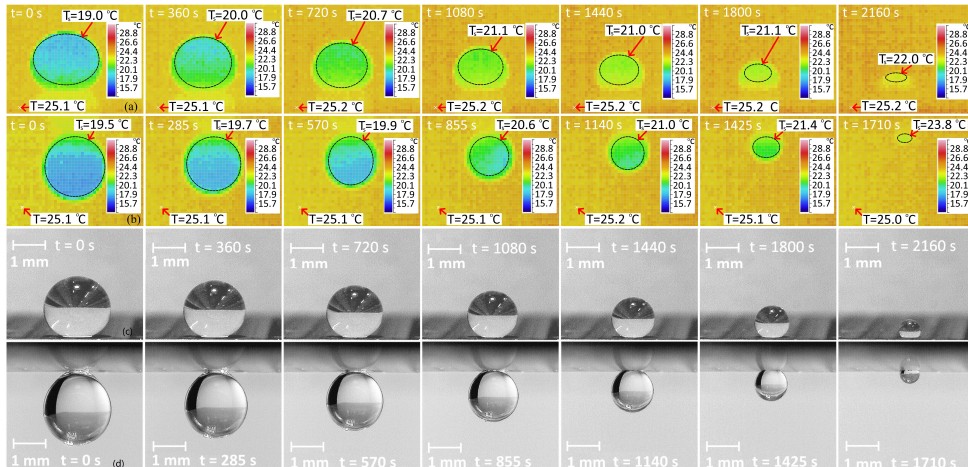

**Figure 3.** Thermal images of droplets: (**a**) sessile; (**b**) pendant. Change in the shape of evaporating droplets: (**c**) sessile; (**d**) pendant.

## 3. Results and Discussion

A comparison of the droplet shapes for the same relative time moment $t/t_{evap}$, where $t_{evap}$ is the full evaporation time of 2400 s and 1800 s for the corresponding sessile and pendant droplets is shown in Figure 4a. The initial droplet diameters differed by ~3%. These data are presented for the case of the evaporation of the suspended and sessile droplets from the same seat so the surface preparation conditions could not affect the process. Usually, the evaporation of droplets on superhydrophobic substrates is accompanied by the instability of the contact line dynamics [52]. As can be seen in Figure 4a, this effect was not observed in the case of the suspended droplet and was absent in the initial stage of the evaporation of the sessile droplet during evaporation on biphilic surfaces. The sessile droplet had a large area of contact with the surface (base diameter). This effect may be due to the fact that the droplet moving under its own weight spread over a large area near the seat so the size of the coffee ring did not coincide with the size of the seat, as shown in Figure 1b.

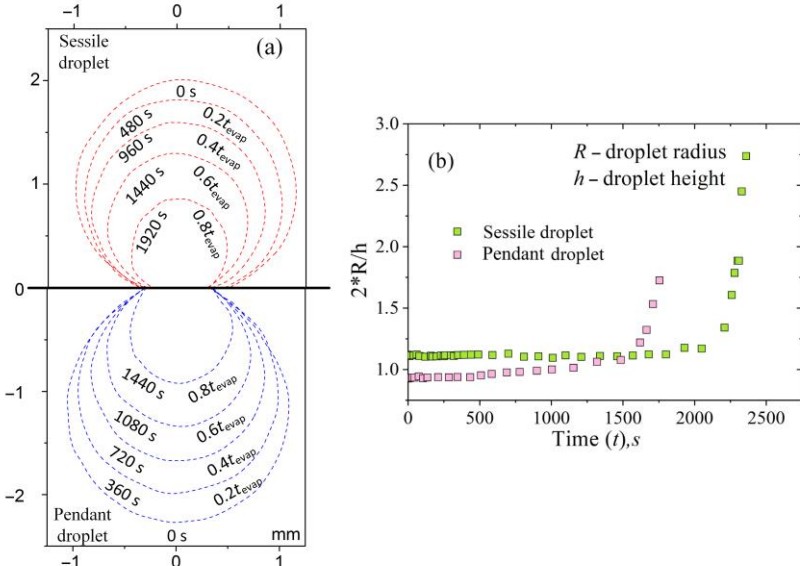

**Figure 4.** (**a**) Contours of sessile and suspended droplets at different times. (**b**) The ratio of the droplet diameter to its height in the process of evaporation of suspended and sessile droplets.

Figure 4b shows that the droplet height decreased almost linearly, regardless of the method of suspension. However, the height of the suspended droplet decreased much

faster (Figure 4b). It should be noted that the droplet height is an important parameter that determines the degree of influence of certain forces acting on evaporating droplets.

A more detailed comparison of the evaporation dynamics of the suspended and sessile droplets is shown in Figure 5. The droplet height, base diameter, and contact angles were analyzed (Figure 5a,c). A comparison of the temperature dynamics of the sessile and suspended droplets is shown in Figure 5b.

For the sessile droplets, the changes in the geometric characteristics can be divided into several regions that are consistent with the temperature data (Figure 5a,b). In the first stage (time interval from 0 to 1440 s), the droplet base diameter changed linearly, the contact angle remained constant and close to the value of the fluoropolymer coating, and the droplet height decreased at a constant rate. The droplet base diameter in this stage was noticeably higher than the superhydrophilic area on the surface. For the size used in the work, the Bond number was quite large ~0.2 (boundary value) at the considered period of time. Thus, the sessile droplet was deformed under the influence of gravity. We need to point out that the fluoropolymer coating was superhydrophobic so the direct contact of the droplet with the fluoropolymer was excluded (we suggest that the superhydrophobic surface was wetted in the Wenzel state). As a result, the sessile droplet hung over the "true" contact area determined by the pinning to the area with superhydrophilic properties; in other words, the "true" contact line was hidden by the side wall of the droplet in the initial stage of evaporation. As the liquid mass decreased, this effect was leveled and the registered contact diameters for the sessile and suspended droplets became almost the same (Figure 5a,c after 1440 s and 1080 s).

In the initial stage of evaporation, a sharp decrease in the surface temperature of the water droplet to 19 °C was observed, after which the droplet temperature changed slightly within a couple of minutes (interval from 0 to 170 s). Further, a gradual increase in the temperature of the droplet in the interval up to 750 s was observed. In the second stage from 750 to 1440 s, the size of the contact line decreased, whereas the contact angle remained constant (depinning mode) and the droplet height decreased at the same rate as in the first stage. In the third time interval from 1440 to 1950 s, a decrease in the droplet height and a decrease in the contact angle and base diameter were observed; this evaporation mode is called the mixed mode. The interval from 750 to 1950 s corresponded to a plateau in the temperature vs. the time graph. In the last stage of evaporation after 1950 s, the contact angle decreased sharply and the height of the droplet changed at a higher rate than in the previous stages, whereas the size of the droplet base remained unchanged. In this time interval, a sharp increase in the droplet temperature to the ambient air temperature was recorded. Similar temperature dynamics were observed in [53].

In the case of the suspended droplet, a smaller number of characteristic regimes in the evaporation dynamics was observed. The diameter of the contact line remained almost unchanged during the entire evaporation process (Figure 5c). In the first stage from 0 to 1080 s, the droplet height decreased at a constant rate, whereas the diameter of the contact line and the contact angle remained constant over time.

As in the case of the sessile droplet, the temperature decreased sharply at the initial time and then gradually increased as the droplet decreased. In the second interval from 1080 to 1440 s, there was a slight change in the diameter of the droplet base and the contact angle, whereas the droplet height decreased at a constant rate. It should be noted that an increase in temperature was observed from 1080 s; the transition to exponential growth was more pronounced at 1440 s. In the last stage of evaporation from 1440 s to the complete drying of the droplet, the contact angle decreased significantly. In the same time interval, the height of the droplet decreased much faster, whereas its base remained constant. The evaporation rate was greater in the last stages than in the initial ones since the minimum film thickness ensured high thermal conductivity. The evaporation of the droplets from a surface with a contact angle of less than 90° was not uniform. That is, the evaporating material flow did not emanate from the droplet as from a sphere (uniformly in all directions)

but diverged along the edges of the droplet [41]. This can also affect the rate of evaporation and deposition processes.

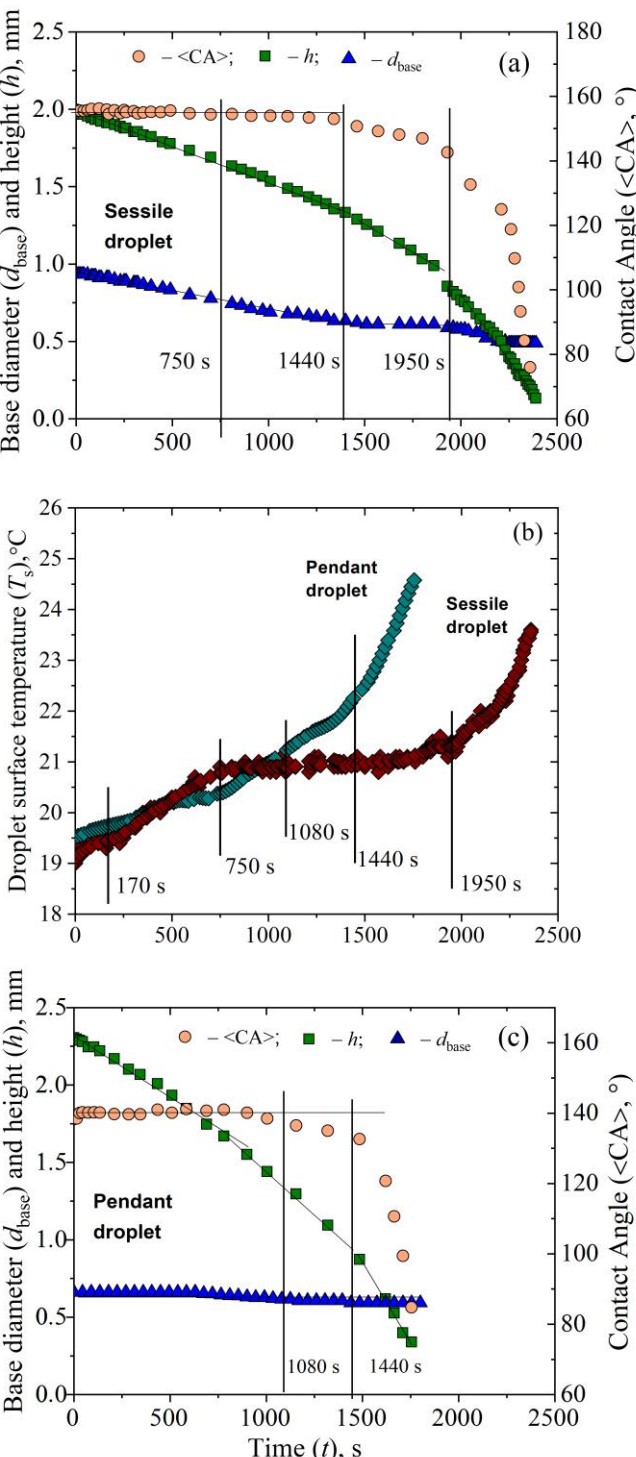

**Figure 5.** Change in the base diameter, height, and contact angle for sessile (**a**) and pendant (**c**) droplets. (**b**) Comparison of the surface temperature dynamics for sessile and suspended droplets.

One of the most interesting aspects of this work was the higher rate of evaporation of the pendant drop, which was accompanied by a higher temperature on its surface. It is well known that in the case of a free droplet, the more intense evaporation leads to a decrease in its surface temperature up to adiabatic temperatures. In this case, this was not observed. Possible explanations are (1) the intense heat flux from the side of the

substrate and (2) the difference in the form of the droplets. As shown in Figure 4a, the main geometric characteristics of the droplets in dimensionless times had similar behavior, and differences were observed only in the surface area of the droplet and the diameter of the base (Figure 5). The surface areas of the droplets were quite close, however, it is unlikely that this significantly contributed to the differences in the evaporation dynamics. This is in agreement with the results of [54], which showed that pendant droplets evaporate slightly slower than sessile droplets in the case of CA = 90°, i.e., the effect of hiding the contact line discussed above did not occur.

The dynamics of the base diameter, unlike the other geometric parameters, were very different for the pendant and sessile droplets. Note that when reaching $0.6t_{evap}$ (these were ~1080 and ~1440 s for the pendant and sessile droplets, respectively), the base diameters became close. From these times onward, the temperature dependence (Figure 5b) and rapid heating of the droplets that were similar in dynamics began. At the earlier times, there was a dramatic difference in the base diameter (as noticed above, it exceeded the pinning region for the sessile droplet). It is known that the main evaporation occurs at the contact line [2]. It was expected that for the sessile droplet, the contact line would make a noticeably smaller contribution to evaporation since it was hidden by an overhanging droplet. Vapors were trapped between the droplet surfaces and the non-wetted wall. This made it possible to more efficiently distribute heat from the substrate over the volume of the droplet and, thereby, increase the evaporation flux from its surface, reducing the average temperature. For the pendant droplet, on the contrary, a lot of liquid evaporated from the contact line and the incoming heat was spent on this process, whereas the rest of the volume did not receive heat from the wall. As a result, the vapor flow from the rest of the droplet area decreased and its average temperature counterintuitively increased. In our experiments, we could not trace the convective flows inside the droplet and the diffusion flows of the vapor outside. A full explanation of the observed effect could be provided by detailed modeling of the processes.

A comparison of the evaporation dynamics of sessile and suspended liquid droplets has not been given much attention in the literature. The rate of evaporation of sessile and suspended droplets of methyl acetoacetate on a smooth surface was studied by Picknett and Bexon [27]. For the smooth surface, these authors used an equiconvex lens with a radius of curvature of 200 mm coated with a 50 μm thick PTFE sticky layer to keep the drop on the inverted surface and provide a sufficiently large contact angle. Due to its small size, the droplet had almost the same shape as the corresponding sessile droplet and the evaporation rate was almost the same. Due to the small initial droplet sizes, the study [27] did not observe the effect of hiding the contact line, which also agrees with our results at the later times of >0.8 $t_{evap}$.

## 4. Conclusions

The shape and size of a droplet placed on a substrate depend on the contact angle, surface tension, substrate tilt, fixation (pinning or depinning) of the line of contact between the droplet and the substrate, etc. Droplet evaporation from biphilic surfaces is of great practical and scientific interest for a deeper understanding of heterophase processes such as wetting, adsorption, precipitation, coagulation, coating, etc. The paper presents a study of the heat and mass transfer during the evaporation of a water droplet from a biphilic surface with different orientations of the droplet relative to the gravitational forces. It should be noted that it is difficult to assess the effect of gravitational force on the rate of droplet evaporation. A feature of this study is the use of a unique approach to the placement and fixation of a droplet on a surface, which made it possible to evaluate the effect of droplet orientation on the evaporation dynamics. The results of the study show that the unique surface structure allows the evaporation of suspended droplets simultaneously in the CCR and CCA modes. An interesting effect was found when comparing the total evaporation time of sessile and suspended droplets: the suspended droplets evaporated 30% faster

and, at the same time, had a higher temperature than the sessile droplets. The temperature evolution of sessile and suspended droplets has a stagewise character.

**Author Contributions:** Conceptualization, M.-K.L.; methodology, V.T. and Y.L.; validation, S.S.; investigation, N.M. and E.S.; resources, S.S. and A.S.; data curation, S.S. and A.S.; writing—original draft preparation, E.S.; writing—review and editing, S.S. and E.S.; visualization, N.M.; supervision, S.S.; project administration, E.S. All authors have read and agreed to the published version of the manuscript.

**Funding:** The investigation of droplet evaporation was supported by the Ministry of Science and Higher Education of the Russian Federation (mega-grant No. 075-15-2021-575). Creating biphilic surfaces was performed with the financial support of the RFBR and NSFC, project number No. 21-52-53025 GFEN_a.

**Institutional Review Board Statement:** Not applicable.

**Informed Consent Statement:** Not applicable.

**Data Availability Statement:** On request.

**Conflicts of Interest:** The authors declare no conflict of interest.

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
