# Peer review of "Evaporation Dynamics of Sessile and Suspended Almost-Spherical Droplets from a Biphilic Surface"

_water, doi:10.3390/w15020273_

Round 1

Reviewer 1 Report

This is an interesting paper describing the results of experimental studies presented in Russia and China. Suggestions to improve the presentation of the results are are made in the attached file. A paragraph giving a brief overview of the modelling approaches would be helpful, but I do not insist on this.

Author Response

Thank you very much for your recommendations. We are glad that you appreciated our work. The necessary corrections were carried out. It allows us significantly improve the manuscript.

Reviewer 2 Report

The authors created a new biphilic surface to study droplet evaporation dynamics and performed good experiments in comparison to show the different performance between a sessile droplet and a pendant droplet.  The excellent data visualization is conveys a clear message and supports the conclusion.  However, some information on experimental control and repeatability is missing.  This work can be published after minor revision. 

Comments:

1.     Overall experiments & Figure 5a & 5c:  Please explain the difference in droplet base diameter for sessile and pendant droplet, and also report experimental control (e.g., sample fabrication variation, measurement/experiment repeatability, measurement accuracy or resolution).  The contact angle dropping while the base diameter remaining the same is the strong evidence that the droplet is pinned at the wettability shift region.  If it’s not the sample fabrication variation for the two samples, then some fundamental understanding is missing for the two scenarios. 

                                    Reference:         Raj et al., Langmuir 2012, 28, 45, 15777–15788

2.     Page 2, line 82-85:  There are several works uses biphilic surfaces, from microscale to macroscale, to characterize wetting behavior.  Also some enhanced boiling studies includes surface wettability.  See above reference and Gao and McCarthy, Langmuir 2009, 25(24)

3.     Page 3, line 123-125:  It’d be better to include the Teflon layer in the heat transfer calculation.  Although the thickness of Teflon is several orders of magnitude lower, the thermal conductivity of Teflon is also orders of magnitude lower than silicon.  Taking 0.3 W/m`K as a nominal value, the thermal resistance is still around 5% of the silicon substrate.  Also, the thickness characterization of the Teflon layer needs to be specified. 

4.     Page 5, Figure 4a:  Please report initial static contact angle of a sessile droplet and a pendant droplet at the same size, and the repeatability of the contact angle measurement.  Generally, gravity effect is not taken into consideration when drop diameter is less than capillary length.  Is it possible that any other parameters are contributing to the variation in contact angle (micro‑roughness, local hydrophobicity from fabrication, etc.)?

5.     Page 5, Figure 4b:  Why are the transition time different for sessile (~2100s) and pendant droplets (~1500s)? 

6.     Page 6, line 225-240:  The author provided some discussion on thermal analysis but does not seem to be convincing enough.  For example, line 226-228 “the heat from the substrate must travel a larger distance through the narrower region of the droplet base to reach the evaporating surface” is a little misleading, since the following parts also discussed different regions and local vapor pressure.  It’d better to rephrase this paragraph to avoid any confusion. 

7.     Page 7, Figure 5b:  The two droplets geometry factors are almost the same from 1000s to 1500s based on Figure 4b, but Figure 5b shows a large difference on surface temperature in this region.  Please explain the reason. 

Author Response

Thank you very much for your recommendations. Please see the attachment.
